# *Drosophila*, an Integrative Model to Study the Features of Muscle Stem Cells in Development and Regeneration

**DOI:** 10.3390/cells10082112

**Published:** 2021-08-17

**Authors:** Hadi Boukhatmi

**Affiliations:** Institut de Génétique et Développement de Rennes (IGDR), Université de Rennes 1, CNRS, UMR6290, 35065 Rennes, France; hadi.boukhatmi@univ-rennes1.fr

**Keywords:** muscle stem cells, satellite cells, *Drosophila*, myogenesis, muscle regeneration

## Abstract

Muscle stem cells (MuSCs) are essential for muscle growth, maintenance and repair. Over the past decade, experiments in *Drosophila* have been instrumental in understanding the molecular and cellular mechanisms regulating MuSCs (also known as adult muscle precursors, AMPs) during development. A large number of genetic tools available in fruit flies provides an ideal framework to address new questions which could not be addressed with other model organisms. This review reports the main findings revealed by the study of *Drosophila* AMPs, with a specific focus on how AMPs are specified and properly positioned, how they acquire their identity and which are the environmental cues controlling their behavior and fate. The review also describes the recent identification of the *Drosophila* adult MuSCs that have similar characteristics to vertebrates MuSCs. Integration of the different levels of MuSCs analysis in flies is likely to provide new fundamental knowledge in muscle stem cell biology largely applicable to other systems.

## 1. Introduction

Skeletal musculature is one of the largest organs of the human body, comprising more than 600 muscles that enable body motion [1]. Skeletal muscles are made of multinucleated myofibers and possess the contractile properties to generate forces. The regeneration of adult muscles, being terminally differentiated, relies on a resident population of stem cells called Satellite Cells (MuSCs), first described by A. Mauro [2]. MuSCs reside underneath the basal lamina and, in healthy individuals, they are remarkably efficient to ensure the homeostasis and regeneration of the skeletal muscles [3,4]. 

Muscle repair is a multistep process. Upon muscle damage, MuSCs proliferate, migrate and divide asymmetrically to give rise to new stem cells and myogenic progenitors, which will ultimately differentiate and fuse with each other or with existing fibers to repair the muscle [5,6]. The cellular and molecular mechanisms regulating MuSC biology are presently a hot topic of investigation: both for answering basic stem cell questions and for possible therapeutic use in treating muscle-degenerative diseases [7,8,9,10]. To this end, a multitude of new experimental models is emerging [11,12,13,14,15]. Among them is the fruit fly, *Drosophila*, in which the genetic and cellular control of MuSC during the course of development and throughout adulthood can be efficiently assessed [16,17,18]. 

*Drosophila* has a long history as a genetic model to study myogenesis, both the muscle structure and core myogenic programs being highly conserved between flies and mammals [19,20,21]. *Drosophila* myogenesis proceeds in two distinct waves, leading to the formation of adequate sets of muscles for the stage-specific modes of locomotion. The first wave happens during embryonic development, and it forms the body wall muscles required for larval crawling. This group of muscles undergoes histolysis from larval to adult histolysis, the pupal transition. The second myogenic wave takes place during the metamorphosis (pupal stages) and gives rise to the adult musculature that allows the animal to feed, walk and fly [19]. Adult muscles are formed from a specific population of MuSC, known as adult muscles precursors (AMPs). They are specified in parallel to the first myogenic wave but are set apart and remain undifferentiated during the whole larval development and ultimately differentiate during metamorphosis to form the adult musculature [22,23]. The AMPs share several features with the vertebrates MuSCs [18,24]. Therefore, studying their specification, maintenance and interaction with their environment have provided numerous insights into the process of muscle development. An important recent discovery was the identification of a new population of AMPs, which persist as undifferentiated cells to the adult stage, and has been proposed to represent *Drosophila* adult Satellite Cells [16,17]. This review presents the recent findings on MuSCs in *Drosophila* and discusses the appealing potential of this integrative model. 

## 2. Making a Muscle Stem Cell

### 2.1. Specification and Positioning of the AMPs

In each embryonic abdominal hemi-segment, there are six abdominal AMPs located in dorsal (D-AMP), dorsolateral (DL-AMP), lateral (L-AMP) and ventral (V-AMP) positions (Figure 1A). While abdominal AMPs are closely associated with the larval muscles and nerves [18,20], thoracic AMPs associates with the wing and leg imaginal discs (Figure 1B) [20]. They express markers specific to muscle progenitors such as the basic helix-loop-helix (bHLH) transcription factor Twist (Twi) [25,26]. In the embryo, AMPs are distributed in a stereotyped pattern that determines the final location of the adult muscles they will form (Figure 1A) [26,27]. Abdominal AMPs will form the adult body wall muscles, and thoracic AMPs associated with the wing and leg discs ultimately form the flight and leg muscles, respectively [22]. Thus, the early steps of AMPs specification are critical to building a proper and complete adult musculature. 

AMP patterning is, at least in part, governed by the Hox genes, which are differentially expressed along the anterior–posterior axis of the embryo [29,30]. For example, the gain of *abdominal-A* function converts the thoracic AMPs into abdominals AMPs suggesting that the Hox inputs control the spatial positioning of the AMPs [31]. Cell–cell communication between the abdominal AMPs also affects their positioning. They send out long cellular processes, which follow the peripheral nervous system and form a network of interconnected cells [32]. Ablation of these cellular processes perturbs their patterning and leads to excessive AMPs number, highlighting an additional level of control in their positioning [33]. 

The wing disc AMPs form a large pool of myoblasts, located in the notal part of the disc, underneath the epithelial cells (Figure 1B) [34]. The disc epithelial cells act as transient niche and provide cues governing AMP proliferation, maintenance and positioning. The role of the epithelial cells in localizing the AMPs at the right place was recently characterized by Everetts et al. [35]. This work revealed the contribution of the FGF signaling in guiding the AMPs to their notal localization. While the FGF-family ligands *thisbe* (*ths*) and *pyramus* (*pyr*) are detected in the epithelial cells of the notum, the receptor *heartless* (*htl*) is specifically expressed in the AMPs. Ectopic expression of the FGF ligands either in the pouch region or along the dorso-ventral axis provokes AMPs spreading towards the corresponding regions. Conversely, loss of either FGF-ligands or of Htl resulted in a reduction of AMPs number linked to increased apoptosis. Collectively, these results point to the role of FGF signaling both in localizing the AMPs to the notum region and sustaining their proliferation and survival. 

### 2.2. The Control of the AMPs Diversity

Similar to vertebrate MuSCs, *Drosophila* AMPs are heterogeneous and express different markers [33,36,37,38]. Each abdominal AMP derives from the asymmetric division of a muscle progenitor (MP) that gives rise to both an AMP and a skeletal muscle founder cell (FC) [39,40]. While the AMPs remain quiescent, the FCs undergo several rounds of fusion with fusion-competent myoblasts to form the larval muscles [41]. The larval muscle shape, size and orientation reflect the early expression of specific combinations of ‘identity’ transcription factors (iTFs) in each FC [40,42]. Hence, every FC has an intrinsic code of iTFs that dictates the properties of the muscle they will form. Likewise, AMPs also differ by the expression of specific iTFs that control their fate and their competence to contribute to different muscle types. One well-studied AMP iTF is Ladybird early (Lbe; the *Drosophila* orthologue of mammalian Lbx1), which is expressed and required for the specification of the lateral abdominal AMPs (L-AMPs) [33]. Lbe is also involved in instructing the identity of leg-associated AMPs by dictating the shape, the structure and the functional properties of the leg muscles deriving from this population [43]. 

Wing disc-associated AMPs form two different types of adult flight muscles: the fibrillar indirect flight muscles (IFMs) and the tubular direct flight muscles (DFMs) [19,20]. These muscles have distinct physiologies, size, contractile properties and thus provided a well-suited system to study the mechanisms behind the early MuSCs divergence during development. The contribution of iTFs in such a divergence was first reported by Sudarsan et al. [34]. The studies identified two pioneer iTFs; Vestigial (Vg) and Cut (Ct), and showed that they are differentially expressed in the wing disc AMPs. AMPs expressing high levels of Vg and low levels of Ct form the IFMs, while the other AMPs (high Ct, no Vg) are required for the formation of DFMs (Figure 1B). Vg is activated by the Wingless signal emanating from the adjacent notal epithelial cells. Vg activates the expression of *spalt-major* (*salm*), a zinc finger TF [44]. Salm is a master regulator of the muscle fibrillar fate, which activates IFM-specific genes and repress genes involved in tubular muscle formation. Consistently, the ectopic expression of Salm in developing leg muscles is sufficient to switch their fate from a tubular to fibrillar organization [44,45]. Interestingly, the morphology of the flight muscle mitochondria is also determined by Salm [46]. *salm* expression is regulated by the homeodomain proteins Extradenticle (Exd) and Homothorax (Hth), which contribute to the fibrillar muscle fate [47]. Thus, Vg, Salm, Exd and Hth transcriptional cascades specify the IFM fate by promoting the expression of the fibrillar-specific genes in AMPs. 

Besides Vg and Ct, little was known about the mechanism that distinguishes between AMPs leading to IFMs versus DFMs. Recent studies using single-cell transcriptomics (scRNA-seq) have considerably improved our understanding of the AMPs diversification [48,49]. Indeed, Zappia MP et al. [50] conducted single-cell RNA-sequencing on wing imaginal discs and showed that AMPs responsible for the formation of IFMs and DFMs have distinct transcriptional signatures and identified new genes differentially expressed between the two populations. Among them, the TF Zfh1 (the *Drosophila* homolog of ZEB1/ZEB2) was found to be highly expressed in the IFMs population, consistently with the dynamics of Zfh1 expression during AMPs specification (Figure 1B) [16]. Zappia et al. have also reported the existence of a large set of new DFM-specific genes, including *kirre*, *midline* (*mid*) and *tenascin accessory* (*ten-a*). Moreover, each of the two populations of AMPs is intrinsically heterogeneous and can be clustered in subpopulations (e.g., according to expression levels of the Notch target genes) that represent various states of myoblasts differentiation. Further analysis of one gene shed in light by this work, *amalgam* (*ama*), which encodes a membrane receptor, showed that its inactivation causes severe muscle phenotypes. This work thus provides compelling evidence that scRNA-seq can identify genes differentially transcribed in the two AMPs populations, while the specific functional requirement of these genes in the process of AMPs diversity remains to be fully characterized. 

When and how can AMPs initiate and maintain a specific transcriptional program? It has been shown that extrinsic signals emanating from the disc epithelium are important for patterning [34]. Wingless signaling that specifies the IFMs lineage is produced from the epithelial wing disc cells closely associated with IFM AMPs, which maintain high levels of Vg [34]. This regulation involves the importin Moleskin (Msk) that regulates the Wingless effector β-catenin/Armadillo (Arm) by controlling its stability and/or nuclear transport [51,52]. Conversely, the Hedgehog (Hh) pathway is required for the specification of DFM AMPs [35]. The Hh ligand is produced by a subset of posterior epithelial cells in close proximity to the AMPs. Although components of the Hedgehog pathway *smoothened* (*smo*) and *cubitus interruptus* (*ci*) are uniformly expressed in most AMPs, the *patched* (*ptc*) receptor is restricted to a subset of DFMs AMPs. *Ptc* expression expanded through the majority of AMPs when exogenous Hedgehog activity was supplied. Reciprocally, the reduction of *smo* levels in AMPs was sufficient to abolish *ptc* expression and to induce defects in adult DFMs. This work suggests that the specification of the DFMs AMPs is controlled by Hh signals emanating from the epithelial cells and supports the view that the microenvironment is pivotal in establishing the AMPs diversity. 

## 3. Role of the Microenvironment in Muscle Stem Cell Maintenance and Activation

### 3.1. Connecting to the Muscles; ‘Homing Behavior’

Once specified and positioned in the right place, abdominal AMPs lie dormant during embryogenesis until the beginning of the larval life. To investigate how AMPs are maintained in a dormant state, Aradhya et al. [53] generated an AMP-sensor line (m6-gapGFP) that enables the visualization of cell shape changes and behavior of the AMPs during development. As described previously [33], embryonic AMPs send out long protrusions and form a network of interconnected cells. In addition, Aradhya et al. showed that AMPs produce numerous smaller filopodia tightly associated with neighboring muscles. The interconnecting cellular processes regulate the maintenance of AMPs since their ablation pushes the AMPs to proliferate prematurely. These connections persist until the first instar larval stage, and they are lost in the second instar larvae. The short filopodia remain, however, associated with the muscles, illustrating that, as with a vertebrate’s satellite cells [54], *Drosophila* AMPs display a homing behavior, likely necessary to sense and respond to instructive signals provided by the muscle fibers. 

### 3.2. Muscle-Driven Insulin Signal Reactivates Dormant AMPs

At the mid-second larval instar, the AMPs are reactivated, exit the quiescent state and enter proliferation to provide the myoblasts that form the adult muscles. The important question is to understand what regulates the transition from a quiescent to an activated state. Aradhya et al. [53] showed that the AMP reactivation is driven by the neighboring muscles, which provide insulin-like peptide 6 (dilp6) to activate the insulin pathway in the AMPs. Filopodia of muscle-associated AMPs facilitate reception of the dilp6 signal from the muscle niche. Subsequently, insulin signaling triggers the Notch pathway in a ligand-independent way, involving the ubiquitin ligase Deltex. Genetic epistasis revealed that AMP proliferation is induced by dMyc, acting downstream of Notch. Thus, the AMPs reactivation requires a nutrient-dependent switch that is sensed by cell processes reaching the surrounding muscles [55]. 

### 3.3. Interplay between the AMPs and Motor Neurons 

Soon after their specification, abdominal AMPs exhibit a round shape and are found in the vicinity of motor axons [27,32,33,56]. At later embryonic stages, AMPs elongate and send out long cellular processes that follow the main branches of the peripheral nervous system. Lavergne et al. [56] recently explored further the interactions between AMPs and the navigating motor axons during embryonic development. Using high-resolution imaging, they showed that the AMPs direct their filopodia towards the axons, suggesting that AMP protrusions may play an active role in guiding them. These studies revealed that one motor axon makes the first contact with a dorsolateral AMP (DL-AMP) and then a second contact with a dorsal AMP (D-AMP) before finding its final destination. They further showed that loss or mispositioning of AMPs affects pathfinding, branching and leads to defective muscle innervation. Interestingly the guiding molecules Sidestep and Side IV were found to be specifically expressed in some AMPs, suggesting a putative role in neuron’s pathfinding. This work showed for the first time that the muscle stem cells dynamically interact with the navigating neurons and ultimately contribute to the proper formation of the neuro-muscular system. 

### 3.4. Signals from the Epithelial Tissue Maintain the Undifferentiated AMPs and Promote Their Proliferation

As with abdominal AMPs, the wing-disc-associated AMPs are also specified early during embryogenesis and remain undifferentiated during the embryonic/larval life. After an initial phase of amplification that relies on symmetrical division, AMPs switch to an asymmetric division mode in which they self-renew and generate a post-mitotic myoblast [22]. In both steps, signals are required from the wing disc epithelium, which acts as a transient niche. 

The first wave of AMPs proliferation is dependent on Notch [22]. This pathway relies on cell-to-cell communication and involves the transmembrane proteins Notch receptor and Delta–Serrate–Lag (DSL) family of ligands. Ligand binding provokes the cleavage of the Notch intracellular domain (Nicd), which translocates to the nucleus, associates with DNA binding proteins of the CLS family (CBF1: RBPJ or Su (H) in *Drosophila*), and binds to DNA to regulate gene expression [57]. Notch activation in the AMPs was first suggested to be dependent on Serrate expression in the epithelial cells [22]. In agreement, loss of Serrate function in epithelial tissues reduces the mitotic activity of the AMPs. Conversely, expressing high levels of an active form of Notch in the AMPs increases their number and results in adult flight muscles defects. Recent studies have shown that the epithelial Delta is also instrumental in activating Notch in the AMPs to regulate their proliferation [58]. Interestingly, Delta is highly enriched in a small group of epithelial cells, deprived of Serrate, and proximal to the AMPs population expressing the Notch target gene *E (spl)-m6.* The production of random clones of epithelial cells expressing Delta is capable of inducing *m6-GFP* expression in adjacent AMPs, demonstrating that the Delta–Notch signaling can be transmitted from the epithelial cells to the AMPs [58]. Lineage tracing experiments have further confirmed that *E(spl)-m6* expressing cells contribute to most AMPs [50]. However, whether Delta–Notch activation induces a symmetric versus asymmetric mode of AMPs division remains an open question. In conclusion, these data demonstrate that the epithelial cells activate Notch in the AMPs via both Serrate and Delta ligands to regulate their proliferation. This also raises the intriguing possibility that differential expression of the DSL ligands by the epithelial cells could lead to the activation of different sets of genes within the AMPs population. 

As in vertebrates, the Notch pathway is also required to maintain the AMPs undifferentiated [59,60,61,62,63]. Genome-wide studies have revealed that Notch activates the expression of the TF Twist (Twi) that, in turn, acts as an anti-differentiation signal. In a feed-forward mechanism, Twi then works together with Suppressor of Hairless (Su (H), homolog of RBPJ) and Nicd to regulate a broad spectrum of genes important for maintaining the AMPs undifferentiated [60], including the Zfh1 and Him TFs. The loss of either *zfh1* or *him* leads to premature differentiation of AMPs, a phenotype similar to that of Notch loss of function [16,59,64]. Conversely, both Zfh1 and Him can suppress the premature differentiation of AMPs induced by Mef2 overexpression [16,64]. Both genes are bound by Su (H) and Twist and are upregulated following Notch activation; they have also been reported to transcriptionally repress the differentiation gene Mef2 [65,66,67].

At the late second instar larval stage, AMPs switch to an asymmetric mode of division [22]. This shift is mediated by a secreted epidermal signal, Wingless (Wg), that regulates Numb expression in the AMPs. Numb is segregated in one of the two daughter cells and inhibits Notch signaling. Loss of function of either Wg or Numb leads to a reduction in the mitotic activity of AMPs and affects their asymmetric division [22]. Interestingly, subsets of the wing disc AMPs extend cellular protrusions, termed cytonemes [68,69]. These later transport Delta ligands to activate Notch in the air sac primordium cells (ASPs). The level of Delta–Notch activation in the ASP is adjusted by Wg signaling from the epithelial cells. Importantly, Wg is taken up from the epithelial cells by the AMPs cytonemes and negatively regulates the levels of Delta and thus ASP Notch activity [70]. This exchange of signaling molecules between the epithelial, AMPs and ASPs is necessary to coordinate their development. Thus, the wing disc epithelium acts as a niche and provides different signals to the AMPs to prevent differentiation, regulate their proliferation and synchronize their development with the surrounding tissues. 

During pupal stages, AMPs finally leave their niche, proliferate and migrate as a swarm toward the myotubes targets [71]. During this period, they are maintained in a semi-differentiated state by continuous Notch activation, where each AMP provides the ligand Delta to its neighbors. Notch signaling in the swarming AMPs represses fusion genes and may maintain Zfh1 and Him. By this atypical form of bidirectional Notch activation, the AMPs are kept undifferentiated while migrating and proliferating. Notch activity decays once the swarming AMPs reach the myotubes, switching off the maintenance genes and allowing the fusion and differentiation [72]. 

## 4. *Drosophila*, a New Model to Study Adult MuSCs

*Drosophila* had long been thought to lack adult satellite cells, leading to speculations about how its muscles could withstand the wear and tear of its active lifestyle. Recent progress in the field has allowed the identification of adult MuSCs in flies and showed that these cells are required for the maintenance and repair of adult muscles (Figure 2). 

### 4.1. Characterization of the Drosophila Satellite Cells

The *Drosophila* MuSCs were first described by Chaturvedi et al. [17] within the adult flight muscles (IFMs). Using electron-microscopy, the authors showed that MuSCs are intercalated between the membrane and the extracellular matrix of the mature fiber. *Drosophila* MuSCs are kept quiescent and, upon injury, they enter proliferation and differentiate to restore the damaged muscle [17,20]. The proliferation of MuSCs after injury relies at least on a Notch–Delta signaling, where Delta is strongly upregulated in the injured fibers. In addition, lineage-tracing approaches have shown that, even in normal conditions, *Drosophila* MuSC provides new differentiated myoblasts to the muscles [16]. Together, these studies thus demonstrate that *Drosophila* MuSCs share morphological and functional features with the vertebrate MuSCs [73]. 

### 4.2. Zfh1/ZEB Maintains Undifferentiated MuSCs: An Evolutionarily Conserved Function

To date, the only known specific marker for *Drosophila* MuSCs is Zfh1/ZEB. ZEB is well known for triggering epithelial-to-mesenchymal transitions (EMT) in both developing embryos and cancer cells [74,75,76]. Zfh1 is also known to regulate mesoderm patterning in the embryo and adult intestinal stem cells [77,78]. Zfh1 is expressed in all AMPs during embryonic/larval development (Figure 1) [67] and persists in the adult MuSCs, where it is required to prevent differentiation and maintain stemness (Figure 2A) [16,17]. Indeed, depleting *zfh1* in the adult MuSCs causes rapid exhaustion of the MuSCs, leading to structural and functional defects of the flight muscles [16]. The identification of Zfh1 as a specific marker of MuSCs has further allowed the generation of various molecular genetic tools (MuSC-specific Gal4 drivers, live reporter lines, etc.), which are key assets to manipulate and track MuSCs in vivo (Figure 2B). However, the gene expression programs acting downstream of Zfh1 to maintain MuSCs remain to be identified. An independent study has shown that ZEB1, an ortholog of Zfh1, is also specifically expressed in the mammalian MuSCs, where it is required to inhibit their myogenic conversion [79]. These pioneering studies indicate that Zfh1/ZEB transcription factors play an evolutionarily conserved role in the maintenance of adult MuSCs. 

### 4.3. Setting Aside MuSCs during Development

In vertebrates, adult MuSCs originate from the pool of embryonic progenitors that will form the skeletal muscles [80,81,82]. Similarly, in *Drosophila*, Chaturvedi et al. showed that MuSCs are lineally descended from the wing disc-associated AMPs [17]. This conclusion was further confirmed by following Zfh1 expression dynamics during the different stages of MuSC specification [16,17]. These findings made *Drosophila* MuSC an ideal system to address the fundamental question of how MuSCs are specified and subsequently protected from differentiation for a prolonged developmental period (Figure 3). Further work has demonstrated that protecting the MuSCs during the pupal stages involves, at least, a switch in *zfh1* RNA-isoforms. Differentiation of the AMPs into functional muscles correlates with expression of the microRNA *miR-8/miR-200*, which targets the major *zfh1-long RNA* isoform and decreases levels of the Zfh1 protein. Upon Notch signaling, a subset of AMPs produces an alternate isoform called *zfh1-short*. The *zfh1* isoform switch implies the selection of an alternate promoter and polyadenylation sites, the latter truncating the 3′UTR so that it lacks the seed sites for *miR-8* regulation. The Zfh1 protein is thus specifically maintained in these cells, enabling them to escape differentiation and persist as satellite cells in the adult (Figure 3). The mechanisms selecting alternate promoters to transcribe specific isoforms remain poorly understood. Looping between promoters and polyadenylation sites may be involved in such regulation, as described in other systems [83,84]. Preferential activation of a specific RNA isoform, with differential sensitivity to microRNAs, is a powerful mechanism for maintaining MuSCs and may be of widespread significance for other types of adult stem cells. The sensitivity of *zfh1/ZEB* to *miR-8/miR-200* microRNAs is evolutionarily conserved across animal species and well known to regulate different developmental and tumorigenic processes [74,85]. 

## 5. Conclusions and Prospects 

The characterization of *Drosophila* AMPs provided an excellent experimental model for dissecting the cellular and molecular mechanisms that govern MuSC in the course of development. One important finding was that the AMPs are capable of modifying their cellular morphology and exchange signals with their local environment, at a long range, to coordinate their development and behavior [56,86]. Interestingly, AMPs also contribute to the establishment of the neuromuscular system by guiding the motoneurons during their pathfinding. Finally, a subset of AMPs escapes the differentiation program and forms the adult MuSCs, which appear highly similar to vertebrate satellite cells. 

Although MuSCs have been extensively studied since their discovery in the early sixties, major gaps still exist in our understanding of their behavior during the early steps of their activation, prior to differentiation. Intravital imaging has proven to be a powerful approach to address such questions [87,88,89]. However, the use of this method in mice models has been constrained both by the complexity of the muscle environment and the long duration of regeneration, which precludes extended live imaging. Therefore, most studies on MuSCs dynamics have relied on in vitro or ex vivo experimental systems [90]. Such approaches are unlikely to fully recapitulate the MuSCs environment, which includes other cell types such as fibro/adipogenic progenitors (FAPs) and immune cells [91], which participate in the muscle injury response [92,93]. Thus, elucidating the bidirectional dialog between these different cell types and MuSCs and underlying genetic and molecular substrates needs to be further investigated. Future studies in *Drosophila* will undoubtedly enable addressing this fundamental question, thanks to the recently generated MuSC-specific tools and the advances in live imaging [94,95,96,97]. Furthermore, during the muscle repair, the progression of MuSCs from quiescence to an activated state is orchestrated by signals provided by the microenvironment (e.g., Notch) [92,98]. Thus, there is a strong need for a better understanding of how MuSCs sense and respond to the signals during the multistep process of their activation. The availability of a fast-growing collection of reporter lines, including live sensors of signaling pathways, will allow conducting such studies with unprecedented resolution [58,99,100]. To conclude, the insights gained on processes regulating the *Drosophila* MuSC should provide new fundamental knowledge, which could contribute to stem-cell-based therapies needed to restore skeletal muscle function in humans when this process is failing [7,101,102]. 

## Figures and Tables

**Figure 1 cells-10-02112-f001:**
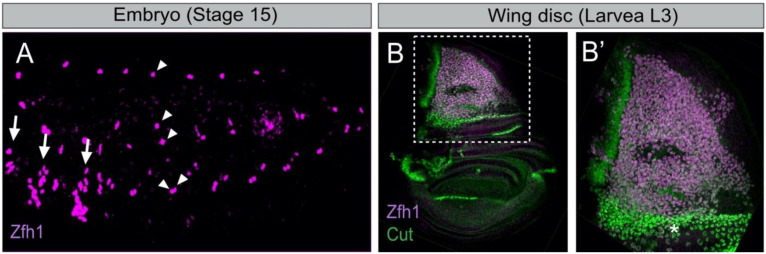
The *Drosophila* embryonic and wing disc associated muscle stem cells express the transcription factor Zfh1. (**A**) Stage 15 embryo stained for Zfh1 (magenta). Arrows and arrowheads indicate the thoracic and abdominal AMPs, respectively (adapted from [28]). The ventral abdominal AMP (V-AMP) is not shown in this sample. (**B**) Zfh1 (magenta) and Cut (green) are expressed in the wing disc AMPs. (**B’**) Higher magnification of boxed region in B shows that high Cut expressing AMPs (asterisk) have low levels of Zfh1 (adapted from [16]).

**Figure 2 cells-10-02112-f002:**
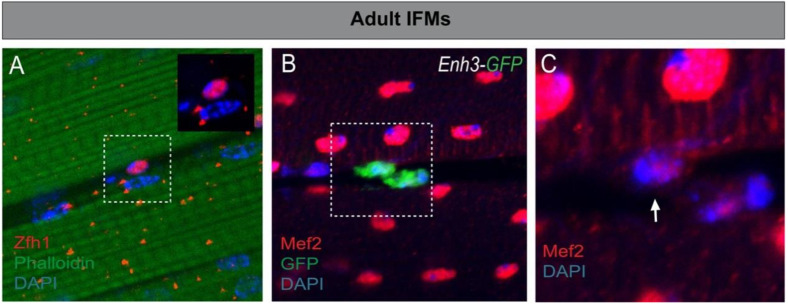
*Drosophila* adult MuSCs. (**A**) Zfh1 (Red) marks the MuSCs associated with the indirect flight muscles (Phalloidin, green), Nuclei (blue). (**B**) *zfh1* enhancer (*Enh3-GFP*, [16]) is activated in the MuSC, characterized by low levels of Mef2 (red, arrow in (**C**) Nuclei (blue).

**Figure 3 cells-10-02112-f003:**
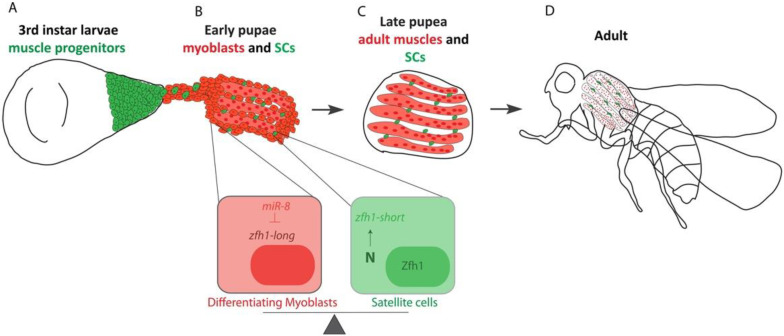
Schematic representation of *Drosophila* MuSC specification during the flight muscle formation (IFMs). (**A**) The adult muscle progenitors (AMPs) of the IFMs are associated with the wing imaginal discs. (**B**) During pupariation, the AMPs migrate toward the DLMs templates. Silencing of *zfh1-long* by *miR-8* (red) facilitates the AMPs differentiation. *zfh1-short* (green) transcription is driven and maintained in MuSCs by Notch signaling. (**C**) At late pupal stages, the MuSCs and the IFMs formation is completed. Since *zfh1-short* is insensitive to *miR-8*, Zfh1 protein is maintained in MuSCs, enabling them to persist in the adult (**D**).

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
