# Peer review of "Drosophila, an Integrative Model to Study the Features of Muscle Stem Cells in Development and Regeneration"

_cells, 2021, doi:10.3390/cells10082112_

Round 1

Reviewer 1 Report

This review provides a high quality and extensive summary of recent advances in the field of Drosophila muscle stem cells and highlights the many advantages of this model for studying homeostasis, the pathways necessary for their reactivation and their involvement in adult muscle repair.

Minor revision:

In figure 1A, are shown with arrowheads 4/5 AMP in an abdominal hemi-segment while the author mentions that there are 6. Could the author replace this figure with the image of an embryo also showing ventral AMPs so that it matches the text better?

Author Response

We are glad that the reviewer finds that our review is relevant for the field and we thank him/her for the comments.

We appreciate the concern of the reviewer about the Figure 1. Given the three-dimensional positioning/distribution of the AMPs, it was technically challenging for us to image an entire embryo showing all the thoracic and all the abdominal AMPs in the same frame. We have added to the figure an extra arrowhead to point the second L-AMP and mentioned in the figure legend that the ventral AMP is not shown in this sample. 

Reviewer 2 Report

In this submission, Dr. Boukhatmi reviews the role of muscle stem cells (MuSC) on muscle growth and repair and how the model organism Drosophila can play a role in future work on stem cell biology and muscle development/repair.  

This was a well-written and concise review paper that gives a good review of the most recent work in the field of muscle stem cell biology and the role Drosophila can play in future research. This could be a good paper to act as a primer to help get people into the field of muscle development/repair. I don't see any need for major changes to this submission and recommend this paper for submission.

That being said, I have one suggestion: there are many abbreviations within this paper and at times it was hard to follow what the abbreviations stood for. It might be helpful to have a small section of the paper dedicated to detailing what the abbreviations stand for, maybe have a small section on each page on the left-hand side detailing this point. A minor suggestion but one that could help the reader(s), especially one not cognizant of the field, to grasp the concepts being put forth by the author.

Author Response

We appreciate that that the reviewer finds our review well-written and provides an up-to-date view of the most recent work in the field of Drosophila muscle stem cells.

We thank the reviewer for the suggestion. We have now included a table summarizing a list of the abbreviations used in the review.